# Disease-associated mutations in the human TRPM3 render the channel overactive via two distinct mechanisms

**Siyuan Zhao, Yevgen Yudin, Tibor Rohacs***

Department of Pharmacology, Physiology and Neuroscience, New Jersey Medical School, Rutgers University, Newark, United States

**Abstract** Transient Receptor Potential Melastatin 3 (TRPM3) is a $Ca^{2+}$ permeable non-selective cation channel activated by heat and chemical agonists such as pregnenolone sulfate and CIM0216. TRPM3 mutations in humans were recently reported to be associated with intellectual disability and epilepsy; the functional effects of those mutations, however, were not reported. Here, we show that both disease-associated mutations in the human TRPM3 render the channel overactive, but likely via different mechanisms. The Val to Met substitution in the S4-S5 loop induced a larger increase in basal activity and agonist sensitivity at room temperature than the Pro to Gln substitution in the extracellular segment of S6. In contrast, heat activation was increased more by the S6 mutant than by the S4-S5 segment mutant. Both mutants were inhibited by the TRPM3 antagonist primidone, suggesting a potential therapeutic intervention to treat this disease.

## Introduction

Transient Receptor Potential Melastatin 3 (TRPM3) is a $Ca^{2+}$ permeable, non-selective cation channel activated by heat (*Vriens et al., 2011*) and chemical activators such as the neurosteroid pregnenolone sulfate (PregS) (*Wagner et al., 2008*) and the synthetic compound CIM0216 (*Held et al., 2015*). TRPM3 is a well-established temperature sensor in peripheral sensory neurons of the dorsal root ganglia (DRG); its genetic deletion in mice leads to defects in noxious heat sensation as well as reduced inflammatory heat hyperalgesia (*Vriens et al., 2011; Vandewauw et al., 2018*). Inhibitors of TRPM3 also reduced both acute heat sensation and inflammatory heat hyperalgesia (*Straub et al., 2013; Krügel et al., 2017*). While TRPM3[-/-] mice show defects in noxious heat sensation, the channel shows increased activity well below the noxious range, when temperature is increased from 15°C to 26°C, with further increases in activity at 37°C (*Vriens et al., 2011*). TRPM3 is also inhibited by activation of Gi-coupled receptors such as μ-opioid receptors and $GABA_B$ receptors in DRG neurons, and agonists of those receptors reduced nocifensive reactions to local injection of TRPM3 agonists (*Badheka et al., 2017; Dembla et al., 2017; Quallo et al., 2017*).

TRPM3 is also expressed in tissues other than peripheral sensory neurons, where its functional roles are not well understood. In pancreatic β-cells, it was shown that application of the TRPM3 agonist PregS induced an increase in insulin secretion (*Wagner et al., 2008*), but TRPM3[-/-] mice showed no impairment in glucose homeostasis (*Vriens et al., 2011*). TRPM3 is also expressed in vascular smooth muscle cells, and PregS was reported to induce contractile responses in freshly isolated aorta, but the concentration required for this response was higher than the plasma levels of this compound (*Naylor et al., 2010*). TRPM3 is also expressed in various regions of the brain, where its functional role is essentially unexplored (*Oberwinkler and Philipp, 2014*).

Very little is known about the physiological and pathophysiological roles of TRPM3 in humans. A recent paper showed that two missense mutations in TRPM3 are associated with a neurodevelopmental disorder with intellectual disability, hypotonia and epilepsy, pointing to important roles of

*For correspondence:
rohacsti@njms.rutgers.edu

**Competing interests:** The authors declare that no competing interests exist.

**eLife digest** Inherited brain disorders often cause severe problems for those affected by them. One example is a group of diseases, collectively termed "developmental and epileptic encephalopathies", or DEE for short. People with these diseases usually have both epilepsy and intellectual disabilities, and in some patients these conditions are associated with two mutations that change a gene called *TRPM3*.

The *TRPM3* gene encodes a protein called an ion channel. Ion channels form pores on the surfaces of cells. When channels are active, the pores open, allowing charged particles – which, in the case of TRPM3, are sodium and calcium ions – to pass through, carrying tiny electrical currents. In the nervous system, ion channels help nerve cells communicate and also allow them to sense changes in the environment.

The TRPM3 channel is known to open in response to heat and certain chemical "activators". In mice, TRPM3 is found in sensory nerve cells, where it acts as a heat sensor. Although altering TRPM3 in mice affects their ability to sense intense or painful heat stimuli, they are otherwise completely normal and have no symptoms resembling human DEE disorders.

Although TRPM3 is found in the human brain, little is known about its role there or what effects the DEE-associated mutations have on its activity. Zhao et al. therefore set out to determine, whether each of the mutation was a 'loss of function', meaning that it stopped the channel from opening, or a 'gain of function', meaning it made the channel open more often.

Frog egg cells and mammalian cells grown in the laboratory were engineered to produce the TRPM3 ion channel. Measurements of electrical activity on these cells revealed that the two mutations seen in people with DEE were both 'gain of function'. Both mutants were more sensitive to heat and chemical activators than the normal protein. They were also more active overall, even without any stimuli. However, one mutation had a greater effect on heat sensitivity, while the other caused a larger increase in chemical-induced activity.

Imaging experiments revealed that both mutant channels also increased the amount of calcium inside the cells. This could explain why the mutations cause disease, since abnormally high calcium levels can damage nerve cells. In addition, the epilepsy drug primidone switched off the mutant channels, pointing to potential treatment of this disease using primidone.

this channel in the human brain (*Dyment et al., 2019*). Seven of the eight patients had a de novo Val to Met substitution in the S4-S5 loop, while one patient had a Pro to Gln substitution in the extracellular segment of S6. The effects of these mutations on channel function however were not reported.

Here, we tested the functional effects of the two disease-associated mutations using electrophysiology and intracellular $Ca^{2+}$ measurements. We find that both disease-associated mutations render the channel overactive. Both mutants showed constitutive activity that was inhibited by the TRPM3 antagonist primidone. As primidone is a clinically used medication (*Krügel et al., 2017*), our findings offer potential clinical intervention to treat this channelopathy. We also find that the Val to Met substitution in the S4-S5 loop induced a larger left shift in the concentration response relationship to PregS and CIM0216 than the Pro to Gln substitution close to the pore-loop in the extracellular segment of S6. The increase in heat activation on the other hand was more pronounced in the S6 mutant. We conclude that both reported mutants of TRPM3 are gain of function, but the mechanism of increased channel activity is different for the two mutants.

## Results

Here we used, patch clamp electrophysiology and intracellular $Ca^{2+}$ measurements in HEK293 cells and two-electrode voltage clamp electrophysiology in *Xenopus* oocytes to study the effects of disease-associated mutations on TRPM3 function. TRPM3 has a large number of splice variants (*Oberwinkler and Philipp, 2014*), but there is no information available which splice variants are expressed in the human brain, and relatively little is known about the functional differences between splice variants. To ensure that our results do not only apply to one variant, we used two commonly used and well-characterized splice variants of TRPM3 in our experiments. In HEK293 cells, we

expressed the human orthologue of the most studied mouse variant TRPM3α2 that was originally cloned from mouse brain (*Oberwinkler et al., 2005*; *Vriens et al., 2011*). In *Xenopus* oocytes, we expressed the human TRPM3 splice variant we used in several previous studies (*Badheka et al., 2015*; *Badheka et al., 2017*), originally described in *Grimm et al. (2003)* also called the TRPM3$_{1325}$ variant (*Oberwinkler et al., 2005*).

Most alternatively spliced exons are in the cytoplasmic N-terminus; thus, the numbering of mutated residues varies between splice variants. The more common S4-S5 segment mutant described as V837M (*Dyment et al., 2019*) corresponds to V990M in the hTRPM3 splice variant we expressed in *Xenopus* oocytes, and to V992M in the hTRPM3α2 variant we expressed in HEK293 cells. The S6 mutant P937Q in Dyment et al corresponds to P1090Q in the hTRPM3 variant we used in oocytes, and it is P1092Q in the hTRPM3α2 we used in HEK293 cells. *Figure 1—figure supplement 1* shows the location of these residues in the TRPM3 sequence and putative structure.

First, we co-expressed the $Ca^{2+}$ indicator GCaMP6f (*Chen et al., 2013*) and the hTRPM3α2 isoform and its mutants in HEK293 cells and performed intracellular $Ca^{2+}$ measurements in a 96-well plate reader. We found that the concentration-response relationship to PregS (*Figure 1A–D*) and CIM0216 (*Figure 1E–H*) were left shifted in the mutant channels; V992M showed a much larger shift than P1092Q for both agonists. The V992M mutant also showed a larger increase in basal $Ca^{2+}$ levels than P1092Q at room temperature (21°C), and the application of the TRPM3 antagonist primidone decreased basal $Ca^{2+}$ levels for both mutants in a concentration-dependent manner (*Figure 1I–L*). Primidone had no effect on basal cytoplasmic $Ca^{2+}$ levels in cells transfected with the wild type TRPM3 indicating negligible basal activity of the wild type channel at room temperature (*Figure 1I*). Primidone robustly inhibited $Ca^{2+}$ signals evoked by $EC_{50}$ concentrations of PregS both for wild type and mutant channels (*Figure 1—figure supplement 2*). We also measured PregS responses at 37°C. Consistent with earlier results (*Vriens et al., 2011*), sensitivity of wild-type TRPM3 to PregS increased at 37°C; the $EC_{50}$ of activation decreased to 0.99 µM at 37°C compared to 7 µM at room temperature (*Figure 1—figure supplement 3A,B*). Basal $Ca^{2+}$ levels at 37°C were substantially elevated in cells expressing wild-type channels, which is consistent with the low temperature threshold of TRPM3 (*Vriens et al., 2011*). Both mutant channels showed very high basal $Ca^{2+}$ levels, which were not further increased by PregS, indicating substantial $Ca^{2+}$ overload when kept at 37°C continuously (*Figure 1—figure supplement 3C,D*). Primidone (50 µM) reduced basal $Ca^{2+}$ levels at 37°C in wild type channels and to a smaller extent in the mutant channels (*Figure 1M–P*).

Next, we transfected HEK293 cells with the mutant and wild-type hTRPM3α2 and used fura-2 $Ca^{2+}$ imaging to study the effects of acutely increased temperatures (*Figure 2*). We first increased the temperature to 37°C, followed by 10 µM primidone at room temperature to facilitate return of $Ca^{2+}$ to baseline. Then we applied 25 µM PregS (in the absence of primidone), and compared the $Ca^{2+}$ responses induced by 37°C to that induced by PregS. In cells transfected with wild-type TRPM3, the temperature-induced $Ca^{2+}$ response was, on average, 29.4% of that induced by PregS, for the V992M mutant it was 76.5%, whereas for the P1092Q mutant it was ~122.5% (*Figure 2D,E*). In cells not expressing TRPM3 increasing temperature to 37°C induced only negligible $Ca^{2+}$ signals (not shown).

Cytoplasmic $Ca^{2+}$ is an indirect measure of TRPM3 activity, thus next, we performed whole cell patch clamp experiments to compare currents induced by increased temperatures and by PregS. These measurements were performed in the absence of extracellular $Ca^{2+}$ to avoid indirect effects of increased cytoplasmic $Ca^{2+}$ such as $Ca^{2+}$ induced desensitization. We stimulated each cell with a temperature ramp from 23°C to 36°C followed by a saturating concentration of PregS (100 µM) at room temperature (*Figure 3A–C*). Since the current amplitudes induced by PregS were highly variable, presumably due to different expression levels of the channel (*Figure 3H*), we normalized the currents induced by increased temperatures to those evoked by PregS and plotted these relative currents as a function of temperature (*Figure 3D–F*). Figure G shows that the slope of the increase of these currents as a function of temperature were significantly steeper for the P1092Q mutant than for V992M, and both mutants were significantly steeper than the wild type.

Next, we expressed the wild-type hTRPM3$_{1325}$ and its V990M and P1090Q mutants in *Xenopus* oocytes and performed full concentration response measurements with the TRPM3 agonist PregS. Consistent with our $Ca^{2+}$ measurements, the concentration response relationships for PregS were left-shifted for both mutants compared to wild-type, but the effect of the V990M mutation was much more pronounced than that of the P1090Q (*Figure 4A–D*).

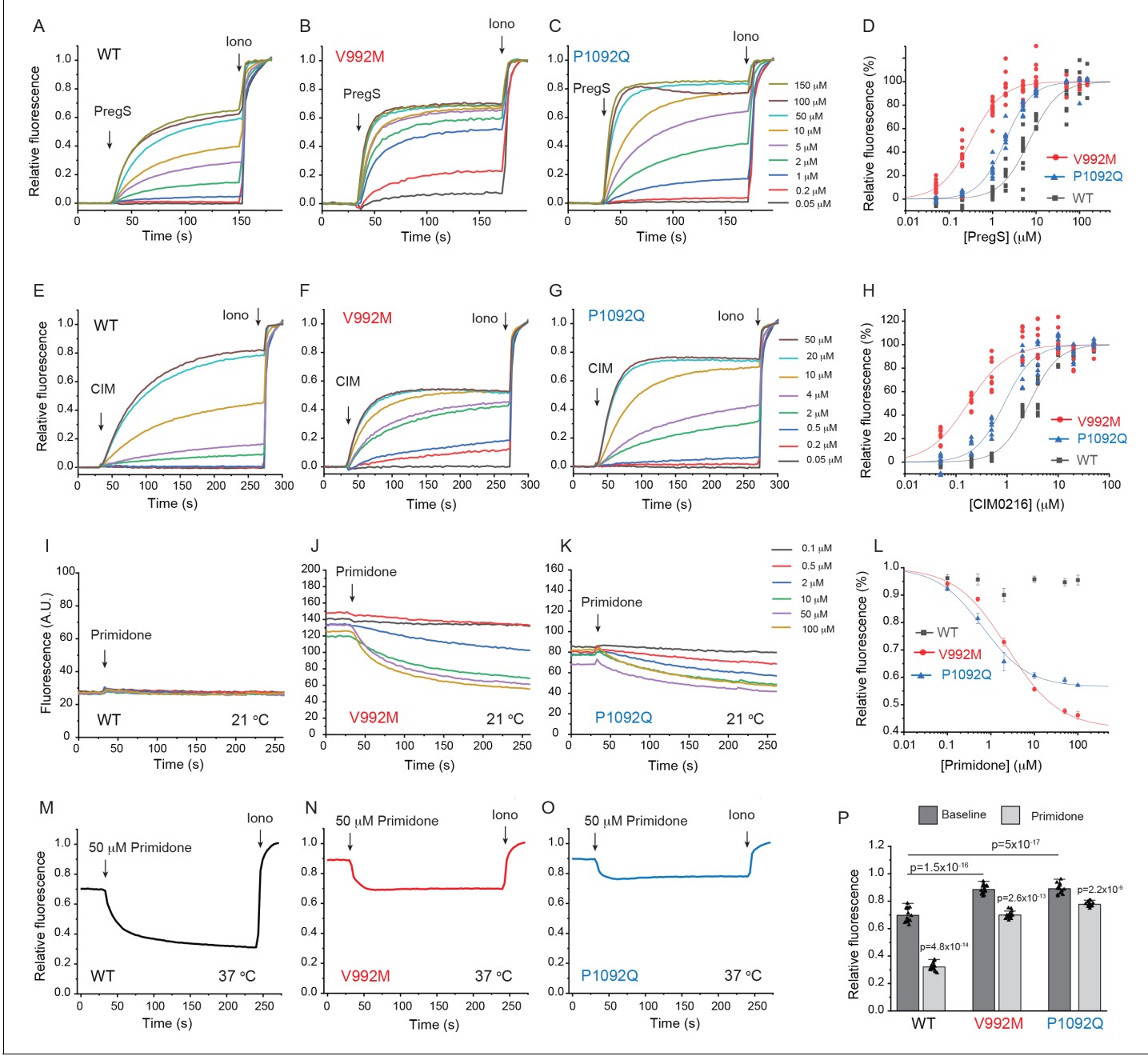

**Figure 1.** Disease-associated mutations increase agonist sensitivity and basal activity of TRPM3. HEK293 cells were transfected with the $Ca^{2+}$ indicator GCaMP6f and the hTRPM3α2 or its mutants, and fluorescence was measured in a 96-well plate reader (Flexstation-3) as described in the Materials and methods section. (**A-C**) Fluorescence traces for TRPM3 (**A**), V992M (**B**) and P1092Q (**C**); the applications of various concentrations of PregS and 2 μM ionomycin are indicated by the arrows. Basal fluorescence before the application of PregS was first subtracted, then the traces were normalized to the fluorescence after the application of ionomycin. Each trace shows the average of four replicates from the same 96-well plate. Measurements were performed at 21˚C. (**D**) Hill fits of the concentration dependence of the fluorescence signals evoked by PregS. Symbols represent individual wells from three independent transfections. The $EC_{50}$ values were 7.01 ± 0.69 μM for wild-type channels, 0.32 ± 0.03 μM for V992M and 1.97 ± 0.08 μM for P1092Q. (**E-G**) Fluorescence traces for TRPM3 (**E**), V992M (**F**) and P1092Q (**G**), the applications of various concentrations of CIM0216 and 2 μM ionomycin are indicated by the arrows. Basal fluorescence before the application of CIM0216 was first subtracted, then the traces were normalized to the fluorescence after the application of ionomycin. Measurements were performed at 21˚C. (**H**) Hill fits of the concentration dependence of the fluorescence signals evoked by CIM0216. Symbols represent individual wells from two independent transfections. The $EC_{50}$ values were 2.72 ± 0.17 μM for wild-type channels, 0.17 ± 0.02 μM for V992M and 0.88 ± 0.08 μM for P1092Q. (**I-K**) Fluorescence traces for TRPM3 (**I**), V992M (**J**) and P1092Q (**K**), the applications of various concentrations of primidone are indicated by the arrows; traces were not normalized and shown as arbitrary fluorescence units (A.U.). Measurements were performed at 21˚C. (**L**) Hill1 fits of the concentration dependence of the inhibition evoked by primidone.

*Figure 1 continued on next page*

*Figure 1 continued*

Symbols represent mean ± SEM from two independent transfections, five or six wells in each. The $IC_{50}$ values were 2.41 ± 0.74 µM for V992M and 0.64 ± 0.09 µM for P1092Q. (**M-O**) Fluorescence traces for TRPM3 (**M**), V992M (**N**) and P1092Q (**O**), the applications of 50 µM primidone are indicated by the arrows; traces show the average of 16 wells from two independent transfections, normalized to the effect of ionomycin. Measurements were performed at 37°C. (**P**) Summary of the data, Mean ± SEM and scatter plots. Statistical significance was calculated with one-way analysis of variance with Bonferroni post hoc comparison for differences of basal fluorescence values between mutant and wild-type channels. The effect of primidone in wild type and mutant channel was evaluated with paired t-test; the p values for significance are shown above the bars.

The online version of this article includes the following figure supplement(s) for figure 1:

**Figure supplement 1.** Location of the V990/992 and P1090/1092 residues.
**Figure supplement 2.** Primidone inhibits $Ca^{2+}$ responses induced by $EC_{50}$ concentrations of PregS.
**Figure supplement 3.** PregS responses of wild type and mutant TRPM3 at 37°C.

Both disease-associated mutations are de novo, and all known patients are heterozygous. To mimic heterozygous conditions, we co-injected oocytes with wild type cRNA and either mutant in a 1:1 ratio. The PregS dose response was still markedly left shifted for the V990M:TRPM3 combination, but it was only marginally shifted in the P1090Q:TRPM3 combination compared to WT TRPM3 (*Figure 4E–G*).

To assess basal current levels, we applied 50 µM primidone at room temperature (20–22°C). Consistent with our $Ca^{2+}$ measurements, primidone evoked a significantly larger inhibition of basal activity in the V990M than in the P1090Q mutant both at 100 and −100 mV (*Figure 4—figure supplement 1B-D*). Primidone did not induce any inhibition in non-injected oocytes (not shown,

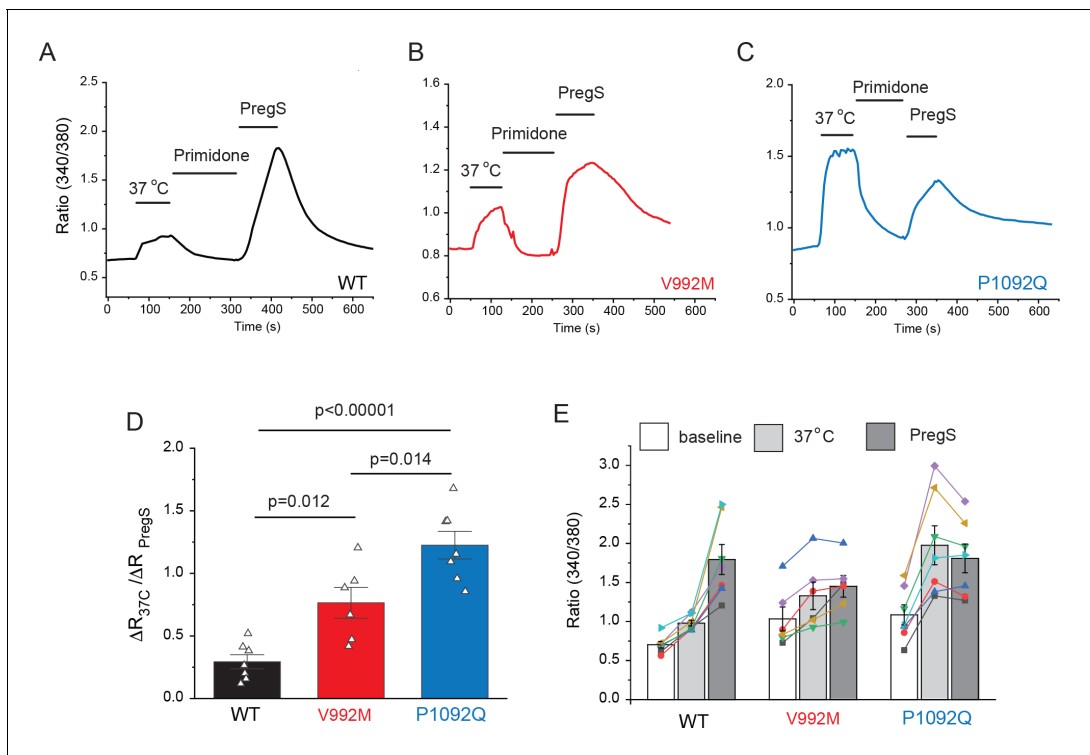

**Figure 2.** Disease-associated mutations increase temperature activation of TRPM3. HEK293 cells were transfected with the human TRPM3α2, or its mutants; fura-2 $Ca^{2+}$ imaging experiments were performed as described in the Materials and methods section. (**A-C**) Averaged fluorescence ratio traces (340 nm/380 nm) for all PregS-responsive cells from one coverslip in cells expressing TRPM3 (**A**), V992M (**B**) and P1092Q (**C**). The cells were first stimulated by increasing the temperature to 37°C, followed by 10 µM primidone at room temperature to facilitate return of $Ca^{2+}$ to baseline, finally 25 µM PregS was applied. (**D**) The change in 340/380 ratio induced by 37°C expressed as a fraction of the response to 25 µM PregS. (**E**) 340/380 ratios at baseline, in response to 37°C and in response to 25 µM PregS. Each symbol represents the average value from all PregS-responsive cells from one coverslip, lines connect data points from the same coverslip. Each coverslips had around 70–80 PregS-responsive cells in the WT group; 35–45 for P1092Q, and 15–25 for V992M. Data were collected from three independent transfections.

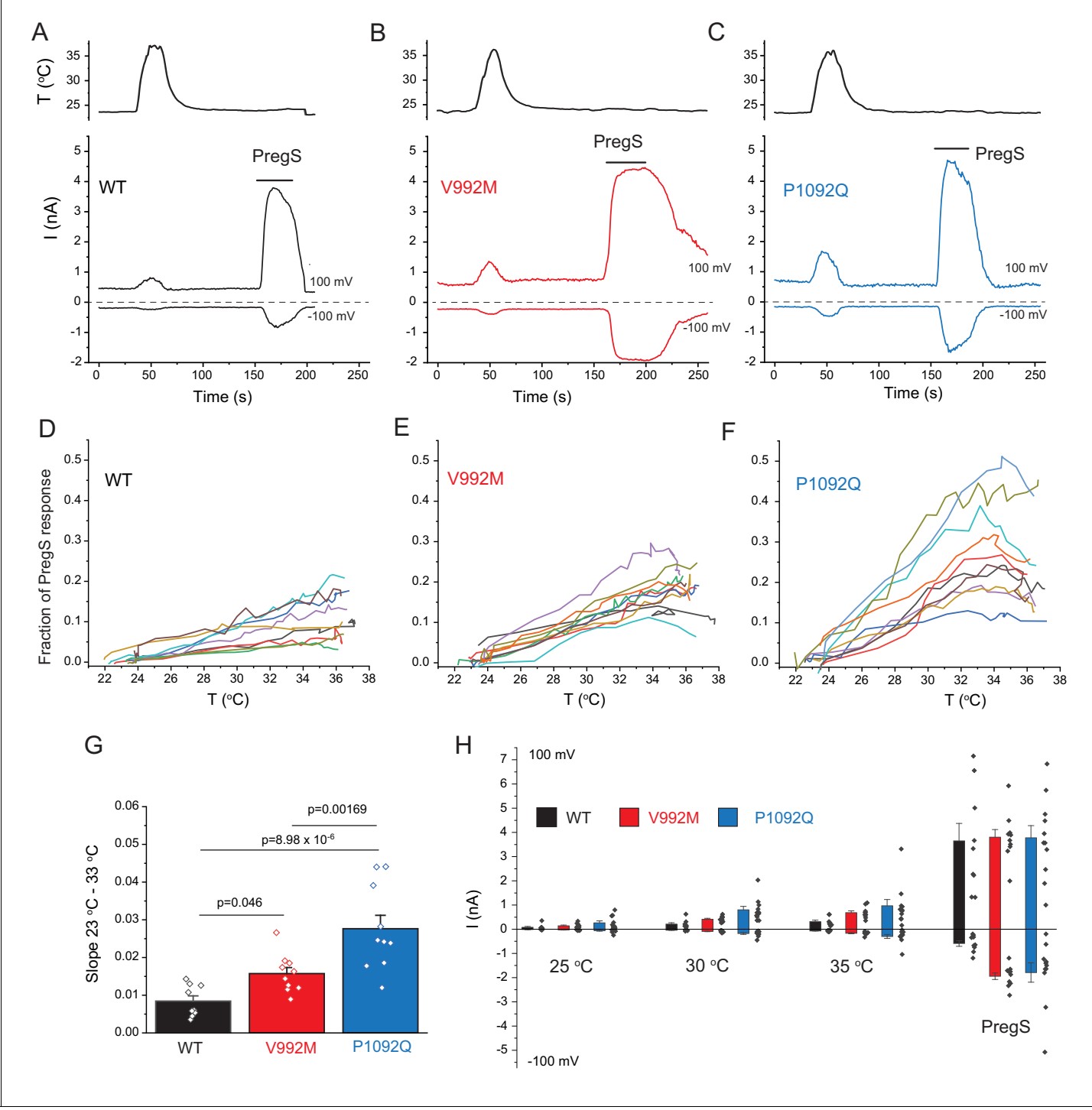

**Figure 3.** Disease-associate mutations increase temperature sensitivity of TRPM3. HEK293 cells were transfected with the human TRPM3α2, or its mutants; whole-cell patch clamp electrophysiology was performed as descried in the Materials and methods section using a ramp protocol from −100 to 100 mV. (**A-C**) Representative measurements, top panels show temperature recordings, bottom panels show currents at 100 mV and −100 mV. The applications of 100 μM PregS are indicated by the horizontal lines. (**D-F**) The heat-induced current amplitudes at 100 mV were normalized to the currents induced by PregS and plotted as a function of the temperature from the same data presented in panels A-C. (**G**) Summary of the slopes of the current increases between 23°C and 33°C determined from linear fits from panels D-F. (**H**) Summary of current amplitudes at 100 and −100 mV induced by increasing the temperature to 25°C, 30°C and 35°C as well as in response to PregS.

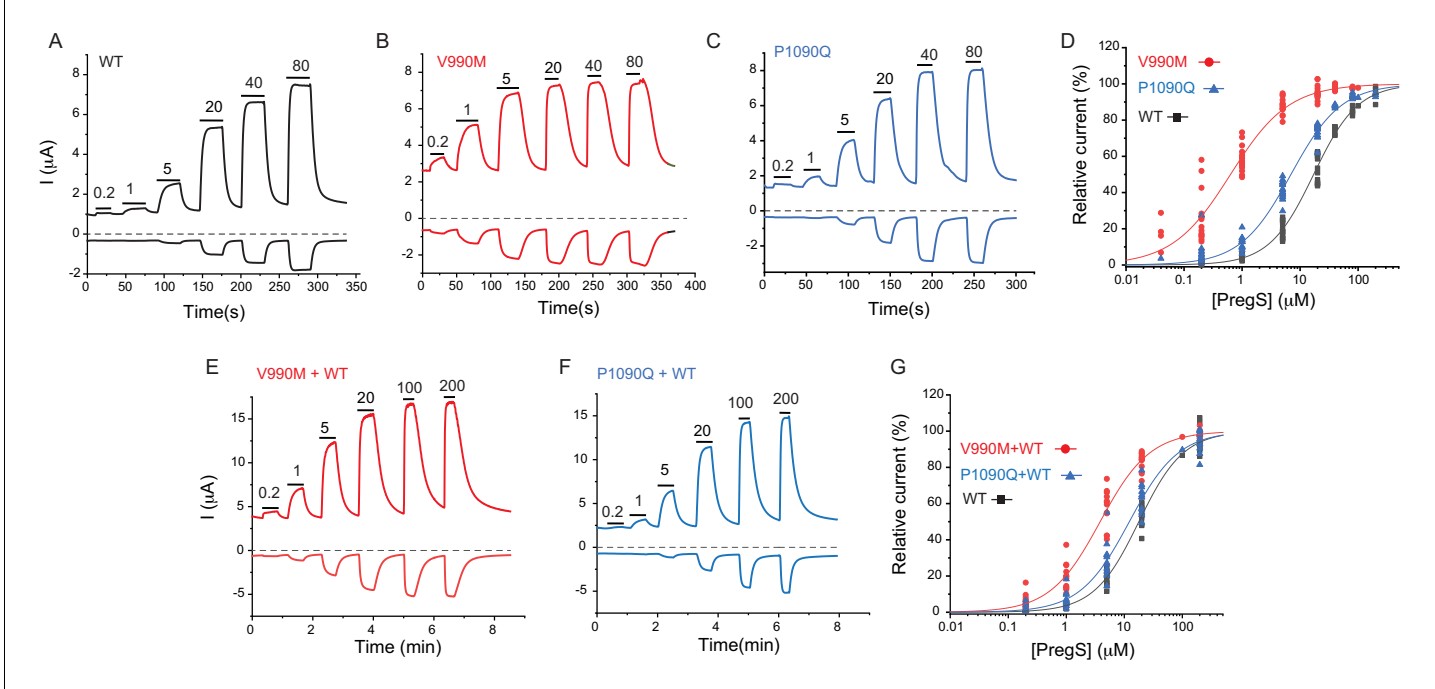

**Figure 4.** Disease-associated mutations increase sensitivity of TRPM3 to PregS. Two electrode voltage clamp experiments in oocytes expressing the hTRPM3$_{1325}$ splice variant and its mutants were performed as described in the Materials and methods section using a ramp protocol from −100 to 100 mV every 0.5 s. (A-C) Representative measurements for wild-type (A), V990M (B) and P1090Q (C); top traces show currents at +100 mV, bottom traces at −100 mV, dashed line shows zero current. The applications of different concentrations of PregS (µM) are indicated by the horizontal lines. (D) Hill fits of the concentration dependence of the effect of PregS at 100 mV for wild-type and mutant channels. The EC$_{50}$ values were 17.1 ± 0.9 µM for wild-type channels, 0.63 ± 0.05 µM for V990M and 6.94 ± 0.38 µM for P1090Q. (E-F) Representative traces for measurements in oocytes injected with cRNA for wild type and mutant channels (1:1 ratio). (G) Hill fits of the concentration dependence of the effects of PregS at 100 mV for wild type and mutant channel combinations; symbols represent individual oocytes from two different preparations. The EC$_{50}$ values were 17.8 ± 1.04 µM for wild-type channels, 3.83 ± 0.27 µM for V990M + WT and 12.2 ± 1.1 µM for P1090Q + WT.

The online version of this article includes the following figure supplement(s) for figure 4:

**Figure supplement 1.** Basal channel activity in disease-associated mutants of TRPM3.

**Figure supplement 2.** Increased temperature activation in disease-associated mutants of TRPM3.

**Figure supplement 3.** M2 muscarinic receptor inhibition of wild type and mutant TRPM3 channels.

n = 5), but it evoked a small reduction in oocytes expressing TRPM3 (*Figure 4—figure supplement 1A,D*). Current amplitudes induced by 100 µM PregS were not different between the two mutants and wild type, at +100 mV, but the V990M mutants showed somewhat larger amplitudes at −100 mV than the wild type or the P1090Q mutant (*Figure 4—figure supplement 1E*). We also compared the basal current amplitudes before applying any stimuli, and found that oocytes expressing the V990M mutant showed higher currents than those expressing P1090Q, and both mutants had higher basal currents than wild-type TRPM3 (*Figure 4—figure supplement 1F*).

We also compared the currents induced by increased temperatures to those evoked by PregS in channels expressed in *Xenopus* oocytes (*Figure 4—figure supplement 2*). We found that the ratio of currents induced by 30°C over those induced by 50 µM PregS were significantly larger in the P1090Q mutant compared to V990M, and both mutants showed significantly larger current ratios than wild type TRPM3. Increasing temperature to 30°C induced negligible currents in non-injected oocytes, but those currents became larger at higher temperatures (not shown), which prevented us from testing higher temperatures in this experimental setting.

TRPM3 has been shown to be inhibited by activation of Gi-coupled receptors via direct binding of Gβγ (*Badheka et al., 2017*; *Dembla et al., 2017*; *Quallo et al., 2017*). To test if the mutations alter receptor-induced inhibition, we co-expressed TRPM3 or its mutants with Gi-coupled muscarinic M2 receptors in *Xenopus* oocytes. *Figure 4—figure supplement 3A–F* show that when we applied acetylcholine (ACh) to stimulate M2 receptors, it evoked a ~ 50% inhibition of TRPM3 currents

induced by 50 µM PregS. The inhibition of the P1090Q mutant was similar to wild type, but the V990M mutant was essentially not inhibited. To test whether the lack of inhibition was due to allosteric effects of the increased sensitivity to PregS, we stimulated the oocytes expressing the V990M mutant with 5 µM PregS where ACh induced a ~ 35% inhibition (*Figure 4—figure supplement 3C, E*). We also tested inhibition at the $EC_{50}$ of PregS for wild type (17 µM) and mutant channels, 0.6 µM for V990M and 7 µM for P1090Q. *Figure 4—figure supplement 3G-K* shows that wild type and V990M mutants were inhibited to a similar extent, the P1090Q mutant was inhibited somewhat more than the wild-type channel. These data show that while the V990M mutation affects receptor-induced inhibition at high PregS concentrations, it is not likely to be the primary mechanism of its gain-of-function phenotype.

## Discussion

Overall, our data show that disease-associated mutations in TRPM3 render the channel overactive. Both mutants showed basal activity even at room temperature, which was reduced by the TRPM3 antagonist primidone. Basal activity of the V992M mutant at room temperature was higher than that of the P1092Q mutant, but at 37°C, the difference in basal activity between the two mutants became negligible. Given the increased constitutive activity of the mutants at body temperature, increased neuronal excitability and/or $Ca^{2+}$-induced neuronal damage is a possible disease-causing mechanism. Primidone is a clinically approved antiepileptic drug; it is thought to exert its effects by being converted to barbiturate by the liver, but it crosses the blood brain barrier (*Nagaki et al., 1999*), and directly inhibits TRPM3 activity even below its therapeutic concentration (*Krügel et al., 2017*). Our data showing that primidone inhibited the basal activity of the mutant channels, suggests a potential therapy for this newly described channelopathy.

The V990/V992M mutation showed a larger increase in basal activity at room temperature, and also induced a larger left shift in the concentration response curves of the agonists PregS and CIM0216 than the P1090/1092Q mutation. This is expected if both mutations increase the stability of the open state of the channel, with the V990/992M mutation having a larger effect. If increased open state stability is the only explanation for the over-activity of the mutants, we would expect V990/992M to be also more sensitive to activation by increased temperatures than P1090/1092Q. This is however not what we observed. We consistently find that the P1090/1092Q mutant showed more pronounced activation by increased temperatures than the V990/992M mutant. This indicates that the mechanism of over-activity is different for the two mutants.

These two residues are in different locations; V990/992 is in the S4-S5 linker whereas P1090/1092 is in the outer portion of S6 (*Figure 1—figure supplement 1*). The S4-S5 linker plays essential roles in channel gating, and has been shown to be a hotspot of disease-associated gain-of-function mutations in a number of TRP channels, including TRPV3, TRPV4, TRPM4 and TRPA1 (*Hofmann et al., 2017*). This channel segment also plays a role in binding of hydrophobic, or amphipathic ligands in TRPM channels (*Huang et al., 2020*). For example, in TRPM8 channels the S4-S5 linker is in direct contact with both the cooling agent icilin, and the menthol analog WS12 (*Yin et al., 2019*). While PregS is thought to activate TRPM3 by directly binding to the channel (*Drews et al., 2014*), its binding site in the channel is not known, and currently there is no structural information available for TRPM3. Because of the clear increase in basal activity of the V990/992M mutant, it is quite likely that the mutation primarily increased the stability of the open state and the decrease in $EC_{50}$ of agonist activation is a consequence of the change in activation equilibrium constant (*Colquhoun, 1998*). It cannot be excluded, however, that the mutation also affected PregS binding concurrently, given the general role of this segment in ligand binding in TRP channels.

The membrane phospholipid phosphatidylinositol 4,5-bisphosphate ($PIP_2$) is also required for PregS-induced TRPM3 activity (*Badheka et al., 2015*; *Tóth et al., 2015*; *Uchida et al., 2016*). In TRPM8, the equivalent of V990 is two positions upstream from a residue that is in close contact with $PIP_2$, and it is 4 and 7 positions downstream from two residues in close contact with icilin and the menthol analog WS12 (*Yin et al., 2019*; *Figure 1—figure supplement 1A*). $PIP_2$ is located adjacent to the menthol analogue WS12 and the cooling agent icilin in the TRPM8 structures, and menthol has been shown to allosterically affect $PIP_2$ activation (*Rohács et al., 2005*). Therefore, it is also possible that the V990/992M mutation affects PregS activation indirectly via $PIP_2$. Increased $Ca^{2+}$ levels

in cells expressing mutated channels may also modify cellular PI(4,5)P$_2$ levels, which may alter channel activity. Differentiating between these possibilities will require future studies.

While the mechanism of temperature activation of TRP channels is not fully understood (*Clapham and Miller, 2011*; *Islas, 2017*; *Arrigoni and Minor, 2018*; *Castillo et al., 2018*), large-scale unbiased mutagenesis studies on TRPV1 (*Grandl et al., 2010*) and TRPV3 (*Grandl et al., 2008*) show that mutations in the pore region and the outer portion of S6 in these channels selectively abolished heat-, but not agonist-induced channel activation. The P1090/1092Q mutation in TRPM3 is located in the outer portion of S6, and it had a stronger effect on heat activation than the V990/992M mutation, therefore it is possible that the primary effect of the P1090/1092Q mutation is increasing heat sensitivity. Increased temperatures synergize with PregS in activating wild-type TRPM3 (*Vriens et al., 2011*), see also *Figure 1—figure supplement 3B*, therefore it is possible that the increased PregS sensitivity of the P1090/1090Q mutant is secondary to its increased heat activation.

Overall our data show that both disease-associated mutations render TRPM3 overactive, but likely with different mechanisms.

# Materials and methods

**Key resources table**

| Reagent type (species) or resource | Designation | Source or reference | Identifiers | Additional information |
|---|---|---|---|---|
| Recombinant DNA reagent | hTRPM3α2; hTRPM3 variant 10 | Genescript | NM_001366141.2 | In pCDNA3.1(+); tagged with GFP on its N-terminus |
| Recombinant DNA reagent | hTRPM3α2-V992M | Genescript | NM_001366141.2 Modified by V992M mutation | In pCDNA3.1(+); tagged with GFP on its N-terminus |
| Recombinant DNA reagent | hTRPM3α2-P1092Q | Genescript | NM_001366141.2 Modified by P1092Q mutation | In pCDNA3.1(+); tagged with GFP on its N-terminus |
| Recombinant DNA reagent | hTRPM3$_{1325}$ | Grimm et al 203 JBC, 278, 21493 | AJ505026 | Original clone was subcloned into the pGEMSH oocyte vector |
| Recombinant DNA reagent | GCaMP6f | Addgene | # 40755 RRID:Addgene_40755 | |
| Cell line (human) | HEK293 | ATCC | CRL-1573, RRID:CVCL_0045 | |
| Strain, strain background | *Xenopus* Leavis | Nasco | LM00535 | Female frogs to extract oocytes |
| Commercial assay or kit | mMessage mMachine | Thermo Fisher Scientific | Catalogue # AM1344 | In vitro transcription kit |
| Commercial assay or kit | Effectene | Qiagen | Catalogue # 301425 | Transfection reagent |
| Commercial assay or kit | QuikChange II XL | Agilent Technologies | Catalogue # 200522 | Site-Directed Mutagenesis Kit |
| Chemical compound, drug | Pregnenolone Sulfate | Cayman Chemicals | Catalogue # 21004 | TRPM3 agonist |
| Chemical compound, drug | CIM0126 | Calbiochem | Catalogue # 534359 | TRPM3 agonist |
| Chemical compound, drug | Primidone | Sigma | Catalogue # p7295 | TRPM3 antagonist |
| Chemical compound, drug | Ionomycin | Cayman Chemicals | Catalogue # 56092-81-0 | Calcium ionophore |
| Chemical compound, drug | Fura2-AM | Invitrogen | Catalogue # F1221 RRID:AB_11156243 | Calcium indicator; |
| Software, algorithm | Origin 2019b | Originlab.com | | |

*Continued on next page*

*Continued*

| Reagent type (species) or resource | Designation | Source or reference | Identifiers | Additional information |
|---|---|---|---|---|
| Software, algorithm | pClamp10.6 | Molecular Devices | RRID:SCR_011323 | |
| Software, algorithm | Image Master 5 | Photon Technology International | | |
| Software, algorithm | Prism6 | www.graphpad.com | | |

## Intracellular Ca$^{2+}$ assay in 96-well plates

Intracellular Ca$^{2+}$ measurements were performed using a Flexstation-3 96-well plate reader with rapid well injection (Molecular Devices) as described earlier (*Hughes et al., 2019*) with some modifications. Briefly Human Embryonic Kidney 293 (HEK293) cells were purchased from American Type Culture Collection (ATCC), Manassas, VA, (catalogue number CRL-1573), RRID:CVCL_0045; cell identity was verified by STR analysis by ATCC. HEK293 cells were cultured in MEM supplemented with 10% FBS and 100 IU/ml penicillin plus 100 µg/ml streptomycin in 5% CO$_2$ at 37°C. Additional cell authentication was not performed, but passage number of the cells was monitored, and cells were used up to passage number 25–30 from purchase, when a new batch of cells was thawed with low passage number; cells were tested for the lack of mycoplasma infection. HEK293 cells were transfected with hTRPM3α2-GFP, or its mutants (200 ng) and GCaMP6 (1 µg) using the Effectene reagent (Qiagen). The human orthologue of the mouse splice variant TRPM3α2 in the pCDNA3.1(+) vector (hTRPM3 variant 10; NM_001366141.2) tagged with GFP on its N terminus, and its V992M and P1092Q mutants were purchased from Genescript, GCaMP6f was a kind gift from Dr. Lawrence Gaspers. After 24 hr, transfected cells were plated on poly-D-lysine coated black-wall clear-bottom 96-well plates and measurements were performed 24–48 hr after plating. Before experiments, the MEM media was replaced with a solution containing (in mM) 137 NaCl, 5 KCl, 1 MgCl$_2$, 2 CaCl$_2$, 10 HEPES and 10 glucose, pH 7.4 and the plate was measured at around 21°C. GCaMP6 signal was measured at excitation wavelengths 485 nm and fluorescence emission was detected at 525 nm. Sampling interval was 0.86 s and four parallel reads were performed for each condition. For most experiments, 2 µM ionomycin was applied to determine the maximum response. Primidone was purchased form Sigma, CIM0216 from Calbiochem, ionomycin and PregS from Cayman Chemicals. For measurements at 37°C the MEM medium used to culture the cells was replaced with the measurement solution preheated to 37°C, and the plate was placed in the plate reader warmed to 37°C using its built-in temperature controller.

## Ca$^{2+}$ imaging experiments

Ca$^{2+}$ imaging experiments were performed using an Olympus IX-51 inverted microscope equipped with a DeltaRAM excitation light source (Photon Technology International, PTI), as described earlier (*Badheka et al., 2017*). HEK293 cells were transfected with hTRPM3α2-GFP or its mutants using the Effectene reagent (Qiagen). Cells were loaded with 1 µM fura-2 AM (Invitrogen) for 40 min before the measurements at 37°C, and dual-excitation images at 340 and 380 nm excitation wavelengths were detected at 510 nm with a Roper Cool-Snap digital CCD camera. Measurements were conducted at room temperature in extracellular solution containing 137 mM NaCl, 5 mM KCl, 1 mM MgCl$_2$, 2 mM CaCl$_2$, 10 mM HEPES and 10 mM glucose, pH 7.4. PregS, and primidone were applied with a gravity-driven whole chamber perfusion system. Temperature stimulation was performed using a custom-built system as described earlier (*Badheka et al., 2017*) by pushing bath solution through a spiral tubing immersed in hot water using a 60 ml syringe while monitoring the temperature of the perfusion chamber using a CL-100 Warner Instruments temperature controller. The analogue signal from the CL-100 unit was fed into the Digidata digitizer and the temperature curve was collected in Clampex. Data analysis was performed using the Image Master 5 software (PTI).

## Whole cell patch clamp experiments

HEK293 cells were transiently transfected with cDNA encoding the hTRPM3α2-GFP, or its mutants with 0.2 µg of constructs using the Effectene reagent (Qiagen) according manufacturer's protocol and were used in experiments 48–72 hr later. Measurements were carried out on GFP positive cells,

in an extracellular solution containing (in mM) 137 NaCl, 5 KCl, 1 MgCl$_2$, 10 HEPES and 10 glucose, pH 7.4. The intracellular solution contained (in mM) 140 potassium gluconate, 5 EGTA, 1 MgCl$_2$, 10 HEPES, and 2 NaATP, pH 7.3. Patch clamp pipettes were prepared from borosilicate glass capillaries (Sutter Instruments) using a P-97 pipette puller (Sutter Instrument) and had a resistance of 2–4 MΩ. In all experiments after formation of GΩ-resistance seals, the whole-cell configuration was established and currents were recorded using a ramp protocol from −100 mV to +100 mV over 500 ms preceded by a −100 mV step for 100 ms; the holding potential was 0 mV, and this protocol was applied once every 1 s. The currents were measured with an Axopatch 200B amplifier, filtered at 5 kHz, and digitized through Digidata 1440A interface. In all experiments, cells that had a passive leak current over 100 pA were discarded. Data were collected and analyzed with the PClamp10.6 (Clampex) acquisition software (Molecular Devices, Sunnyvale, CA), and further analyzed and plotted with Origin 2019b (OrigiLab, Northampton, MA). Heat stimulation was performed as described for the Ca$^{2+}$ imaging experiments.

### Two electrode voltage clamp experiments

*Xenopus laevis* oocytes were prepared as described earlier (*Badheka et al., 2015*). All animal procedures were approved by the Institutional Animal Care and Use Committee at Rutgers New Jersey Medical School. In brief, frogs were anesthetized in 0.25% ethyl 3-aminobenzoate methanesulfonate solution (MS222; Sigma-Aldrich); bags of ovaries were removed surgically from the anesthetized frogs. Individual oocytes were obtained by overnight digestion at 16°C in 0.2–0.3 mg/ml type 1A collagenase (Sigma-Aldrich), dissolved in a solution containing 82.5 mM NaCl, 2 mM KCl, 1 mM MgCl$_2$, and 5 μM HEPES, pH 7.4 (OR2 solution). The next day the collagenase containing solution was discarded and the oocytes were washed multiple times with OR2 solution. The oocytes were maintained in OR2 solution supplemented with 1.8 mM CaCl$_2$, 100 IU/ml penicillin, and 100 μg/ml streptomycin at 16°C. cRNA was transcribed from the linearized human TRPM3 (hTRPM3) cDNA clone (*Grimm et al., 2003*), or its mutants in the pGEMSH vector using the mMessage mMachine kit (Thermo Fisher Scientific). cRNA (40 ng) was microinjected into individual oocytes, using a nanoliter-injector system (Warner Instruments). For combined injection of wild-type and mutant TRPM3 for *Figures 4E–G*, 40 μg total cRNA was injected in a 1:1 ratio. The V990M and P1090Q mutants were generated using the QuikChange II XL Site-Directed Mutagenesis Kit (Agilent Technologies). For the GPCR regulation of these mutants, we injected cRNA of human M2 muscarinic receptors together with TRPM3 or mutants at 1:1 ratio. Oocytes were used for electrophysiological measurements 48–72 hr after microinjection. The hTRPM3$_{1325}$ clone in a mammalian expression vector was provided by C. Harteneck (Eberhard Karls University Tübingen, Tübingen, Germany), and it was subcloned into the pGEMSH oocyte vector using standard molecular biology techniques.

Two electrode voltage clamp experiments were performed as described (*Badheka et al., 2015*). In brief, oocytes were placed in extracellular solution (97 mM NaCl, 2 mM KCl, 1 mM MgCl$_2$, and 5 μM HEPES, pH 7.4), and currents were recorded with thin-wall inner filament– containing glass pipettes (World Precision Instruments) filled with 3 M KCl in 1% agarose. Currents were measured with a ramp protocol from −100 to 100 mV once every 0.5 s with a GeneClamp 500B amplifier and analyzed with the pClamp 9.0 software (Molecular Devices). PregS, ACh and primidone were applied with a gravity driven whole chamber perfusion system. Temperature stimulation was performed the same way as for the whole cell patch clamp and Ca$^{2+}$ imaging experiments in HEK293 cells.

### Statistics

Statistical analysis was performed with Origin 2019b and Prism6. Data are plotted as mean ± SEM and scatter plots. No statistical method was used to predetermine sample sizes, but our sample sizes are similar to those generally employed by the field. Experiments were performed in random order. Data were analyzed with t-test, or one-way analysis of variance with Bonferroni's post hoc test, p values are reported in the figures.

## Acknowledgements

This study was supported by grants NSNS055159, GM131048 and GM131048 to TR. The hTRPM3$_{1325}$ clone in a mammalian expression vector was provided by C Harteneck (Eberhard Karls University Tübingen, Tübingen, Germany), the human M2 muscarinic receptor cDNA in the pGEMHE

vector was a kind gift from Diomedes Logothetis (Northeastern University, Boston, MA). The authors appreciate the help of Dr. Paula Bartlett and Dr. Andrew Thomas with the 96-well plate reader, and the help of Mr. Michael Motto with the plate reader data analysis.

## Additional information

### Funding

| Funder | Grant reference number | Author |
|---|---|---|
| National Institute of Neurological Disorders and Stroke | NS055159 | Tibor Rohacs |
| National Institute of General Medical Sciences | GM093290 | Tibor Rohacs |
| National Institute of General Medical Sciences | GM131048 | Tibor Rohacs |

The funders had no role in study design, data collection and interpretation, or the decision to submit the work for publication.

### Author contributions

Siyuan Zhao, Conceptualization, Data curation, Investigation, Writing - original draft, Writing - review and editing; Yevgen Yudin, Investigation, Writing - review and editing; Tibor Rohacs, Conceptualization, Supervision, Funding acquisition, Visualization, Writing - original draft, Writing - review and editing

### Author ORCIDs

Yevgen Yudin ⬡ https://orcid.org/0000-0002-0663-6435
Tibor Rohacs ⬡ https://orcid.org/0000-0003-3580-2575

### Ethics

Animal experimentation: This study was performed in strict accordance with the recommendations in the Guide for the Care and Use of Laboratory Animals of the National Institutes of Health. All of the animals were handled according to approved institutional animal care and use committee (IACUC) of Rutgers University, protocol number 17024.

### Decision letter and Author response

Decision letter https://doi.org/10.7554/eLife.55634.sa1
Author response https://doi.org/10.7554/eLife.55634.sa2

## Additional files

### Supplementary files

• Transparent reporting form

### Data availability

All data generated or analysed during this study are included in the manuscript and supporting files.

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
