## [Decision Letter]

**Acceptance summary:**

The function of TRPM3 channels in the brain is not well understood. Recently, single-point mutations in this channel were reported to be the substrate of a form of epilepsy and the authors of the present manuscript now provide functional characterization of those mutants. Their data show that the mutations are gain-of-function events, producing hyperactive channels that as a consequence might contribute to increased excitability leading to the pathological phenotype.

**Decision letter after peer review:**

Thank you for submitting your article "Disease-associated mutations in TRPM3 render the channel overactive via two distinct mechanisms" for consideration by *eLife*. Your article has been reviewed by three peer reviewers, including Leon D Islas as the Reviewing Editor and Reviewer #1, and the evaluation has been overseen by a Reviewing Editor and Richard Aldrich as the Senior Editor. The following individuals involved in review of your submission have agreed to reveal their identity: Haoxing Xu (Reviewer #2).

The reviewers have discussed the reviews with one another and the Reviewing Editor has drafted this decision to help you prepare a revised submission.

Summary:

The physiological functions of TRPM3 ion channels are not well understood. TRPM3 has been shown to act as a peripheral nervous system mild temperature heat-sensor. However, the roles it might play in the central nervous system are less well known. Recently, single point mutations have been identified in TRPM3 from human patients with a neurodevelopmental disability and epilepsy. This finding suggests that TRPM3 channels may play a very important physiological role in normal aspects of motor control and cognition.

This manuscript by Zhao at al., presents an initial characterization of the functional effects of these two mutations on the molecular physiology of hTRPM3. The findings show that both mutants produce a gain-of-function phenotype that partially explains the pathological effects of the mutations.

Essential revisions:

1) The authors use two "commonly used" splice variants of the channel but do not specify if these are the variants expressed in the brain in humans. It seems that both of those are not the same splice variant where the mutations were recently found (Dyment et al., 2019). The authors should perform the analysis on the same splice variant that was sequenced in human patients.

2) For the V990M and P1090Q, the authors explain that "temperature sensitivity" was affected (to different degrees), however, they have not measured temperature sensitivity, that is, the slope of the conductance vs. temperature curve. The authors should be careful to distinguish this measurement from the amplitude of the current at a fixed temperature, which is what they are measuring and may or may not have anything to do with changes in sensitivity.

3) For the mutant in the pore (P1090Q), the authors should show if the effect of the mutant is also on the single-channel conductance. This is especially true to begin to distinguish the effects on temperature activation.

4) The effects of the mutations are characterized by comparing magnitude of the current elicited by ligands or temperature with the magnitude of the currents elicited by pregnenolone sulphate (PS), but the authors have not characterized the effects of the mutations on the sensitivity of the channels to PS. It is paramount that the authors show that the mutants have not affected activation by PS since this could severely bias the interpretation of the results.

5) The interpretation that each one of the mutations affects heat or agonist activation preferentially is not well sustained. Without measuring the actual sensitivity to temperature, you cannot argue that it has been changed by the mutation. The same goes for sensitivity to agonist. Without knowing the binding site you cannot make an argument about sensitivity. Given that TRPM3 seems to be a channel that acts by allosterically coupling its different modalities of activation, and the fact that both mutations increase the basal activity of the channel to some degree, it the seems likely that both mutations just increase the basal open probability without actually affecting the sensitivity to temperature or to ligand. Given that these channels are gated by allosteric mechanisms, increasing the basal open probability will be reflected in increased fractions of current being activated by either temperature or ligand binding. Since the authors do not yet have a handle on the detailed effects of the mutations, they should be careful to offer mechanistic molecular explanations.

6) Given that the heat sensitivity of TRPM3 might be relevant for central neurons, you may consider testing the PS sensitivity of the mutant channels at the body temperature of 37°C. Results from such experiments would dramatically boost the significance of the study. In addition, it is important to note that the elevated basal Ca^2+^ levels in the mutant-expressing cells may alter the chemo- and thermos- sensitivities indirectly, e.g., by affecting the levels of PIP2, which is known to regulate TRPM3. This should be discussed.

7) The temperature-dependent gating described here should be thoroughly reviewed. TRPM3 is known to be activated by noxious heat, but in the current study, the channel is robustly activated by 30°C. Judging from the recordings in the Figure 2, it appeared that there was no temperature threshold for activation. Note that the temperature protocol appears to be slightly misaligned with the current traces, such that the currents started to increase prior to temperature elevations. The authors should also consider using the same experimental protocol and X/Y axis to compare WT and mutant channels.

8) In Figure 4, the experimental design is flawed. The effects of drug inhibition should be separated from those caused by temperature reduction (washout).

9) The authors analyze one splice variant in oocytes (using electrophysiology) and the second one in HEK cells using calcium imaging. The authors did not provide any reason for their choices. Also, the authors used primidone only in HEK cells, although they show higher basal activity in oocytes as well, why did the authors not perform parallel experiments for both splice variants?

10) Figure 1: The authors assume that the co-expression of wild-type and mutant receptors (in 1:1 ratio) will lead to heteromeric channels. What is the experimental evidence for this assumption? Moreover, the fact that the mutations are heterozygote, according to Dyment et al., 2019, does not necessarily mean a heteromeric channels. Thus, it is not clear what the advantage is of these experiments.

11) Figure 1—figure supplement 2: According to the present results, the agonist CIM0216 evokes a non-washable response in oocytes in all the concentrations. Thus, establishing a relevant dose-response under these conditions is quite challenging. The authors should perform a single dose recording normalized to the saturating concentration and not a step dose-response. More importantly, the mutant V990M seems to lose all sensitivity to the CIM in negative potentials. Alternatively, the data for CIM should be eliminated.

12) Figure 1—figure supplement 3: The authors measure the Gi evoked TRPM3 inhibition of the wild type and mutants. Although the conclusion that the mutations do not interfere with the Gi inhibition seems correct, the experiment should be done at the relevant EC50 of each construct and not in the wild type saturating concentration.

13) Figure 2: similar to the previous point, the authors should normalize the temperature response to the same PregS saturating level based on their dose-response (Figure 1). Using the saturating dose of the wild-type receptor is not comparable when the authors claim to a leftward shift in the mutant's dose-response.

14) Figure 2—figure supplement 1: The authors should provide some recording examples to these basal currents. Based on Figure 2A-C, it seems all constructs have a similar baseline.

15) Figure 3 and Figure 4: As mention previously, it is not clear what is the purpose of this specific splice variant to this study, and why HEK cells and calcium imaging. This data can be added as a supplement to Figure 1.

---

## [Author Response]

Summary:The physiological functions of TRPM3 ion channels are not well understood. TRPM3 has been shown to act as a peripheral nervous system mild temperature heat-sensor. However, the roles it might play in the central nervous system are less well known. Recently, single point mutations have been identified in TRPM3 from human patients with a neurodevelopmental disability and epilepsy. This finding suggests that TRPM3 channels may play a very important physiological role in normal aspects of motor control and cognition.This manuscript by Zhao at al., presents an initial characterization of the functional effects of these two mutations on the molecular physiology of hTRPM3. The findings show that both mutants produce a gain-of-function phenotype that partially explains the pathological effects of the mutations.Essential revisions:1) The authors use two "commonly used" splice variants of the channel but do not specify if these are the variants expressed in the brain in humans. It seems that both of those are not the same splice variant where the mutations were recently found (Dyment et al., 2019). The authors should perform the analysis on the same splice variant that was sequenced in human patients.

To our knowledge there is no information on which splice variants are expressed in the human brain. These mutations are identified in genomic DNA sequences and therefore are found in all splice variants, as neither mutation is located on an alternatively spliced exon. There are more than 17 splice variants of TRPM3 in mice and a similar number in humans; most of them have not been functionally characterized. The authors identifying these mutations did not perform any functional experiments; rather they chose an arbitrary splice variant as a reference sequence to assign numbers to the mutated residues. We chose the two best characterized splice variants to characterize these mutations, to ensure that the effect of the mutations are not specific to one arbitrarily chosen splice variant. We clarified this in the revised version.

2) For the V990M and P1090Q, the authors explain that "temperature sensitivity" was affected (to different degrees), however, they have not measured temperature sensitivity, that is, the slope of the conductance vs. temperature curve. The authors should be careful to distinguish this measurement from the amplitude of the current at a fixed temperature, which is what they are measuring and may or may not have anything to do with changes in sensitivity.

We have performed new experiments for the revised version, where we measured TRPM3 currents in response to slower temperature ramps, and plotted the current increase as a function of temperature. Our data show that the P1092Q mutant showed a significantly steeper increase in currents than the V992M mutant with increasing temperature, and both mutants showed significantly steeper increase than wild type. The new data are shown in Figure 3 of the revised version.

3) For the mutant in the pore (P1090Q), the authors should show if the effect of the mutant is also on the single-channel conductance. This is especially true to begin to distinguish the effects on temperature activation.

As the major effect of the P1090Q mutant was not increased current amplitudes, but rather increased sensitivity to heat and left shifted concentration response relationship to agonists, measuring single channel conductance would not explain the key effect of the mutation, and we felt that detailed single channel characterization of the mutations is beyond the scope of the manuscript. I would also like to refer to my phone conversation with Dr. Islas who confirmed that this experiment was not critical for the revised version.

4) The effects of the mutations are characterized by comparing magnitude of the current elicited by ligands or temperature with the magnitude of the currents elicited by pregnenolone sulphate (PS), but the authors have not characterized the effects of the mutations on the sensitivity of the channels to PS. It is paramount that the authors show that the mutants have not affected activation by PS since this could severely bias the interpretation of the results.

It is not clear what the reviewers meant, we have performed PregS concentration dependence measurements, which were shown in Figure 1 and Figure 3 and the V990/992M mutation induced a larger left shift than the P1090/1092Q mutation. As the figures have been rearranged, these data are shown in Figure 1 and Figure 4 in the revised version.

5) The interpretation that each one of the mutations affects heat or agonist activation preferentially is not well sustained. Without measuring the actual sensitivity to temperature, you cannot argue that it has been changed by the mutation. The same goes for sensitivity to agonist. Without knowing the binding site you cannot make an argument about sensitivity. Given that TRPM3 seems to be a channel that acts by allosterically coupling its different modalities of activation, and the fact that both mutations increase the basal activity of the channel to some degree, it the seems likely that both mutations just increase the basal open probability without actually affecting the sensitivity to temperature or to ligand. Given that these channels are gated by allosteric mechanisms, increasing the basal open probability will be reflected in increased fractions of current being activated by either temperature or ligand binding. Since the authors do not yet have a handle on the detailed effects of the mutations, they should be careful to offer mechanistic molecular explanations.

We agree with the reviewer that we do not have a full mechanistic explanation for the effect of the mutations; we have substantially rewritten the discussion and tried our best to be more careful with interpreting our data in the revised version. When referring to agonist sensitivity we meant the shift in the concentration response curve, and did not mean ligand binding, for which we agree with the reviewer that we need knowledge of the binding site. Overall, however, we do not believe that a simple increase in basal activity can explain the differences between the functional effects of the two mutations. The V990/992M mutation induced a larger increase in basal activity and a larger left shift in the PregS dose response than the P1090/1092Q mutation. If both mutants simply increased basal activity, i.e. the stability of the open state, then the V990/992M mutation would also have a larger effect on temperature activation than the P1090/1092Q mutation. This is however not the case, rather it is the opposite, the P1090/1092Q mutation had a larger effect on temperature activation than V990/992M. As mentioned earlier, in the revised version we also confirmed this finding with measuring the slope of the current increase which showed that the P1092Q mutant showed a significantly steeper increase in currents with increasing temperatures than V992M (Figure 3).

6) Given that the heat sensitivity of TRPM3 might be relevant for central neurons, you may consider testing the PS sensitivity of the mutant channels at the body temperature of 37°C. Results from such experiments would dramatically boost the significance of the study. In addition, it is important to note that the elevated basal Ca^2+^ levels in the mutant-expressing cells may alter the chemo- and thermos- sensitivities indirectly, e.g., by affecting the levels of PIP2, which is known to regulate TRPM3. This should be discussed.

We have performed additional experiments at 37°C in the plate reader assay. As expected, the EC50 for PregS activation of the wild type channel shifted to the left. In both mutants however basal cytoplasmic Ca^2+^ levels became so high, that PregS did not induce any additional increase in cytoplasmic Ca^2+^. These new data are shown in Figure 1—figure supplement 3. We also applied primidone (50 μM) which decreased basal Ca^2+^ levels in both mutants at 37°C similar to that of the basal Ca^2+^ levels in cells transfected with the wild type channels. Primidone also decreased basal Ca^2+^ levels in cells transfected with wild type TRPM3 at 37°C, which is consistent with our data showing that increasing temperature to 37°C from room temperature induces channel activity. These new data are shown in Figure 1 Panels M-P. We have also discussed the possibility of indirect effects by the mutations via PIP2, and possible effects of increased Ca^2+^ levels on PIP2. The patch clamp and two electrode voltage clamp experiments were performed in the absence of Ca^2+^ therefore Ca^2+^ induced changes in PIP2 levels are unlikely in those experimental settings.

7) The temperature-dependent gating described here should be thoroughly reviewed. TRPM3 is known to be activated by noxious heat, but in the current study, the channel is robustly activated by 30°C. Judging from the recordings in the Figure 2, it appeared that there was no temperature threshold for activation. Note that the temperature protocol appears to be slightly misaligned with the current traces, such that the currents started to increase prior to temperature elevations. The authors should also consider using the same experimental protocol and X/Y axis to compare WT and mutant channels.

Genetic deletion of TRPM3 leads to defects in noxious heat sensations, but the channel is known to be activated by temperature well below noxious temperatures; thus, our data are congruent with the literature. We explained this in the introduction of the revised version. The current increasing earlier than the temperature was a plotting error during file conversion, we fixed it in the revised version, but the reviewer is correct that there do not seem to be a threshold, as soon as the temperature increases the current starts increasing also. This is again consistent with data in the literature, the article first describing temperature activation of TRPM3 noted that TRPM3 currents increase when the recording temperature is increased from 15 to 26^o^C with a further increase of currents at 37^o^C (Vriens et al., 2011).

8) In Figure 4, the experimental design is flawed. The effects of drug inhibition should be separated from those caused by temperature reduction (washout).

The purpose of applying primidone in Figure 4 (Figure 2 in the revised manuscript) was to accelerate return of the Ca^2+^ levels to baseline after increased temperature, not to test the effect of the drug per se. We mentioned in the main text, but we also clarify this in the figure legend.

9) The authors analyze one splice variant in oocytes (using electrophysiology) and the second one in HEK cells using calcium imaging. The authors did not provide any reason for their choices. Also, the authors used primidone only in HEK cells, although they show higher basal activity in oocytes as well, why did the authors not perform parallel experiments for both splice variants?

We have provided a better justification for using two splice variants and performed experiments in *Xenopus oocytes* with primidone. The new data are shown in Figure 4—figure supplement 1A-E.

10) Figure 1: The authors assume that the co-expression of wild-type and mutant receptors (in 1:1 ratio) will lead to heteromeric channels. What is the experimental evidence for this assumption? Moreover, the fact that the mutations are heterozygote, according to Dyment et al., 2019, does not necessarily mean a heteromeric channels. Thus, it is not clear what the advantage is of these experiments.

We did not say the channels were forming heteromers, as we agree with the reviewer that we cannot be certain of that without experimental evidence. We think this experiment is important as it presumable mimics the heterozygous condition, regardless of whether the mutant channels form heteromers. In heterozygous conditions one chromosome has the mutant the other chromosome has the wild type gene, and we think it is a reasonable assumption that they generate mRNA of the wild type and mutant channels in similar amounts. We clarified this in the revised version. Note that in the revised version the figure describing these data became Figure 4.

11) Figure 1—figure supplement 2: According to the present results, the agonist CIM0216 evokes a non-washable response in oocytes in all the concentrations. Thus, establishing a relevant dose-response under these conditions is quite challenging. The authors should perform a single dose recording normalized to the saturating concentration and not a step dose-response. More importantly, the mutant V990M seems to lose all sensitivity to the CIM in negative potentials. Alternatively, the data for CIM should be eliminated.

We have removed the CIM0216 oocyte data from the revised version.

12) Figure 1—figure supplement 3: The authors measure the Gi evoked TRPM3 inhibition of the wild type and mutants. Although the conclusion that the mutations do not interfere with the Gi inhibition seems correct, the experiment should be done at the relevant EC50 of each construct and not in the wild type saturating concentration.

We have performed experiments with M2 receptors using EC50 concentrations of PregS from wild type and mutant channels; the new data are shown in Figure 4—figure supplement 3G-K. The data are consistent with our initial assessment that interfering with Gβγ inhibition is not the major mechanism of overactivity in these mutants.

13) Figure 2: similar to the previous point, the authors should normalize the temperature response to the same PregS saturating level based on their dose-response (Figure 1). Using the saturating dose of the wild-type receptor is not comparable when the authors claim to a leftward shift in the mutant's dose-response.

The oocyte experiments with temperature induced currents are moved to the supplement (Figure 4—figure supplement 2) as the new patch clamp experiments in HEK cells (Figure 3) convey a very similar message. The concentration response curves are left shifted in the mutants, therefore a saturating concentration for the wild type is also saturating for the mutants.

14) Figure 2—figure supplement 1: The authors should provide some recording examples to these basal currents. Based on Figure 2A-C, it seems all constructs have a similar baseline.

This figure was a compilation of data from the original Figure 1, Figure 2 and Figure 1—figure supplement 2, example traces were provided in those figures. We have performed additional experiments with the TRPM3 inhibitor primidone in oocytes, which we show together in the revised manuscript with the summary of basal currents, providing additional representative traces in the same figure (Figure 4—figure supplement 1A-C).

15) Figure 3 and Figure 4: As mention previously, it is not clear what is the purpose of this specific splice variant to this study, and why HEK cells and calcium imaging. This data can be added as a supplement to Figure 1.

There are several reasons we think these experiments are important. 1. The splice variant we used in the original Figure 3 and 4 (Figure 1 and Figure 2 in the revised MS) is the human orthologue of the best characterized mouse splice variant TRPM3a2. TRPM3 has a large number of splice variants (>17), testing the effect of mutations all of them is not realistic, but we wanted to ensure that our data are not only valid on splice variant. 2. The oocyte experiments were performed in the absence of Ca^2+^ to avoid complications with Ca^2+^ induced inactivation. The Ca^2+^ imaging data show that a similar effect is also induced on Ca^2+^ signals evoked by TRPM3 activation. 3. As it was pointed out the CIM0216 dose response in oocytes is difficult to interpret, and our Ca^2+^ measurements show that the CIM0216 dose response is also left shifted. Nevertheless, our new HEK cell electrophysiology experiments (Figure 3) made the original Figure 2 with heat-induced currents in oocyte redundant; therefore, we moved that figure in the supplement (Figure 4—figure supplement 2).